

# Goal adjustment and well-being after an acquired brain injury: the role of cognitive flexibility and personality traits

Gunther Van Bost[1,2], Stefaan Van Damme[1] and Geert Crombez[1]

[1] Department of Experimental Clinical and Health Psychology, Universiteit Gent, Ghent, Belgium
[2] Unit Acquired Brain Injury, Centrum voor Ambulante Revalidatie Ter Kouter, Deinze, Belgium

## ABSTRACT

**Objective:** The tendency to flexibly adjust goals that are hindered by chronic illness is related to indicators of wellbeing. However, cognitive flexibility is often impaired in persons with an acquired brain injury (ABI), possibly affecting the ability to flexibly adjust goals. In this study we examined whether cognitive flexibility is positively related with the ability to disengage from goals to reengage with goals in persons with ABI. Second, we explored whether goal adjustment abilities are predictive of a unique proportion of the variance inabilities are predictive of quality of life and life satisfaction after controlling for personality traits.

**Method:** Seventy-eight persons with an ABI completed a set of questionnaires. Goal disengagement and goal reengagement were assessed using the Wrosch Goal Adjustment Scale (GAS). Indicators of wellbeing were measured with the European Brain Injury Questionnaire (EBIQ) and the Satisfaction with Life Scale (SWLS). The percentage of perseverative errors on the Wisconsin Card Sorting Test (WCST) was used as an indicator of cognitive inflexibility. Big Five personality traits were assessed *via* the NEO Five Factor Inventory (NEO-FFI). Four hierarchical multiple regression analyses were then conducted. The first two analyses tested the effect of cognitive flexibility on goal adjustment tendencies. The second two analyses tested whether goal adjustment has a predictive value for life satisfaction and QOL beyond personality.

**Results:** Cognitive flexibility was positively related to goal reengagement, but not to goal disengagement. Goal reengagement was positively associated with both quality of life and life satisfaction after controlling for demographic, illness characteristics and personality factors. Goal disengagement was negatively related to life satisfaction.

**Conclusion:** Flexible goal adjustment abilities have a unique explanatory value for indicators of wellbeing, beyond personality traits. The findings indicate that in persons with lower cognitive flexibility, goal reengagement ability might be negatively affected, and should be taking into account during rehabilitation.

Corresponding author
Gunther Van Bost,
gunther.vanbost@ugent.be

## INTRODUCTION

An acquired brain injury (ABI) is an injury to the brain that is not hereditary, congenital, degenerative, or induced by birth trauma (*Brain Injury Association, 1997*). Often these injuries are caused by a head traumata or stroke (*Avesani et al., 2013*). Less common causes are anoxia, brain tumours, intoxication and brain infections. Patients with ABI may experience a variety of consequences. Many suffer from problems in mobility and communication (*Lannoo et al., 2004*). Cognitive and behavioural consequences, such as difficulties in attention, memory, planning, and impulse control are less visible but are often more challenging (*Lassaletta, 2019*). Most individuals with ABI experience disability and a loss of quality of life (QOL) (*Mar et al., 2011*). *Jacobsson & Lexell (2013)* found that even 10 years after the injury, life satisfaction of people with an ABI is lower than that of a sex- and age-matched group. For many, an ABI causes a discontinuity in life that has proven hard to accept (*Kim et al., 2019*).

The consequences of an ABI may jeopardize the realisation of important goals (*Doering et al., 2011*). Goals are internal representations of states that a person aims to accomplish by their behavior (*Austin & Vancouver, 1996*). Engagement towards personal goals is important for people's subjective well-being (*Carver & Scheier, 2008*). Unforeseen events, such as a chronic illness, or the gradual decline of capacities because of aging may hinder attainment of these goals. If people persist in their striving towards goals that are no longer attainable, this has a detrimental impact on their well-being (*Kuenemund et al., 2013*). Developed in the context of life-course dynamics, dual process model of *Brandtstädter & Rothermund (2002)* describes how assimilative coping strategies aim at a tenacious goal pursuit, whereas accommodative coping strategies focus on flexible goal adjustment. When confronted with difficulties in goal pursuit people may excert more effort or look for alternative strategies to obtain these goals. When people are confronted with repeated failure to realise their premorbid goals despite extra effort and the use of new strategies people can become more open to alter these goals, setting the stage for an accommodative adaptive respons. Central to this accommodative mode is the flexible adjustment of the goals to the available resources (*Brandtstädter, 2009*).

*Wrosch et al. (2003)* distinguish between two processes relevant for accomodative coping. Goal disengagement means that an individual stops pursuing a specific goal and can let go. This may help people to avoid the frustration of repetitive confrontation with failure. Goal reengagement is the process of finding and engaging in new, more feasible goals, and may offer the satisfaction of fulfilling new meaningful goals. The benefits of goal adjustment strategies on well-being have been shown in the context of different chronic diseases, such as cancer (*Mens & Scheier, 2016*), multiple sclerosis (*Van Damme et al., 2019*), and hearing loss (*Garnefski & Kraaij, 2012*). Similar results have been found in people with an ABI (*Van Bost, Van Damme & Crombez, 2020*). In their review of the literature concerning goal adjustment in people with long-tern health conditions, such as cancer and stroke, *Scobbie et al. (2020)* concluded that goal disengagement and goal reengagement had a positive association with recovery and well-being, although the findings for goal disengagement were mixed. In clinical studies (*Van Bost, Van Damme &*

*Crombez, 2020*; *Van Damme et al., 2019*) goal reengagement was positively associated with subjective QOL and life satisfaction. No effect of goal disengagement was found.

In individuals with ABI goal adjustment can be hampered because of the cognitive impairments related to the brain injury. The process of disengaging from previous goals and reengaging towards new goals may require a certain level of cognitive flexibility. Indeed, both flexible goal adjustment and cognitive flexibility refer to an ability to change behaviour in response to environmental changes. Cognitive flexibility is one of the executive functions (*Lezak et al., 2012*) and includes seeing things from a different perspective, creative thinking, and flexible adapting to changed circumstances (*Diamond, 2013*). Individuals with an ABI may then show an impaired ability to respond to environmental feedback (*McDonald, Flashman & Saykin, 2002*). A clinical standard to assess cognitive flexibility is the Wisconsin Card Sorting Test (WCST, *Gelfo, 2019*). In this test people first must find a rule for sorting cards. Once they have established that, at some point without warning the rule is changed and people need to discover the new rule. In this situation, some people immediately start looking for the new sorting rule, whereas others perseverate in using the previous rule despite repetitively receiving the feedback that their answers were false. Individuals with ABI show less cognitive flexibility on the WCST than that the population norm and this is not limited to people with frontal lesions (*Norup & Barcelo, 2009*). An impaired cognitive flexibility may then also reduce the ability to adjust goals. As far as we know there is no research about the impact of impaired cognitive flexibility on goal adjustment in general and more specifically in individuals with ABI. The present study therefore investigated the relationship between cognitive flexibility and two forms of flexible goal adjustment, namely goal disengagement and goal reengagement.

Another factor that may affect flexible goal adjustment is personality. People vary in their tendency towards the use goal disengagement and goal reengagement strategies. In fact, *Wrosch, Scheier & Miller (2013)* see these tendencies as reflecting an underlying stable trait, influencing people's responses towards unattainable goals across multiple domains and situations. The traitlike character of goal adjustement tendencies raises the question to what extent they have an unique role in explaining well-being beyond the general personality traits. The five-factor model of personality (*Costa & McCrae, 1992*), comprising of Neuroticism, Extraversion, Openness, Agreeableness, and Conscientiousness, is broadly adopted. The relationship between personality traits and well-being in the general population is well-established. In their meta-analysis *Steel, Schmidt & Schultz (2008)* found that the Big Five personality factors could explain between 39% and 64% of the variance in indicators of well-being. Extraversion and Neuroticism are the strongest predictors of life satisfaction (*Schimmack et al., 2004*). *Dwan & Ownsworth (2017)* found that in persons with a stroke a higher Neuroticism was consistently related to a poorer well-being, with the effects of the other personality traits being mixed.

*Carver & Connor-Smith (2010)* state that there is a complex interplay between personality, coping and well-being. Extraversion is linked to more approach tencencies (*Lengua et al., 1999*) and could therefore lead to more goal reengagement. People with a

high degree of neuroticism are more inclined towards avoidance behavior, possibly facilitating goal disengagement. Conscientiousness is associated with a deliberated problem solving approach, which can lead to more goal disengagement towards unattainable goals, as well as to more goal reengagement towards new, more feasible goals. People that are situated higher on certain personality traits may be more or less inclined towards goal reengagement or goal disengagement. Therefore we want to study if goal disengagement and goal reengagement have an unique value beyond the Big Five personality traits in explaining life satisfaction and quality of life of people with an ABI. In summary, this study investigates the relationship between goal disengagement, goal reengagement, and cognitive flexibility and their effects on indicators of well-being in persons with an ABI after controlling for personality. First, we investigate whether the score on a test for cognitive flexibility is positively associated with goal disengagement and goal reengagement. Second, we want to investigate whether goal adjustment strategies have a unique explinatory value for disease specific QOL and life satisfaction after controlling for the Big Five personality traits.

## METHOD

### Participants

Seventy-eight persons (49 male and 29 female) with an ABI participated in this study. The majority of 78 participants was recruited from two outpatient rehabilitation centers, five were recruited from a psychiatric unit specialized in caring for patients with neurological disorders, and five from a specialized private practice of a psychologist. All participants lived in Flanders (the Dutch speaking region in the North of Belgium). A rehabilitation psychologist or occupational therapist from the participating centres asked their patients whether they wanted to participate in this study. Inclusion criteria were: (1) participants were between 18 and 67 years old, (2) they had a nonprogressive ABI of any aetiology confirmed by neurological data, (3) they had sufficient command of the Dutch language, and (4) they are considered to be able to complete questionnaires based on clinical judgement. We excluded people with a high probability of relapse, such as people recovering from a brain tumour, because this prognosis could have a different impact on how they experience their future goals.

Participants' age ranged from 19 to 66 years ($M = 44.38$; $SD = 14.50$). Forty-eight participants suffered a stroke. Twenty-six had a traumatic brain injury, three participants had brain surgery for a benign tumour, and one had an ABI following cardiac arrest. On average participants had 12.76 years of formal education ($SD = 2.69$; 9–19). The time elapsed since the injury ranged from 4 to 295 months ($M = 27.42$, $SD = 40.46$). Forty-nine participants (63%) lived with their partner, eighteen (23%) were single, and eleven (14%) lived with their parents or other relatives.

### Measures

An index of the ABI severity was obtained by asking the responsible rehabilitation professional to provide an expert rating on a seven-point scale, ranging from "perfect age-appropriate functioning" over "minor problems" to "severe impairment".

This professional was an experienced rehabilitation psychologist or an occupational therapist who had worked with the respondent for at least 3 months. We obtained separate scores for motor impairments, the communication impairments, the cognitive impairments and the level of self-awareness (*Van Bost, Van Damme & Crombez, 2017, 2020*).

Goal adjustment was measured using the Dutch version of the Wrosch Goal Adjustment Scale (GAS; *Wrosch et al., 2003*). Each of the ten items are scored on a five-point scale, ranging from completely disagree (1) to completely agree (5). Four items form the Disengagement scale, measuring how easy a person can let go of goals (*e.g.*, "I stay committed to the goal, I can't let go"). The other six items form the Reengagement Scale (*e.g.*, "I think that I have other meaningful goals to pursue"). We found a Cronbach's α of 0.73 for the Disengagement scale and 0.89 for the Reengagement scale.

The Big Five personality factors were measured with the Dutch version (*Hoekstra, Ormel & de Fruyt, 2002*) of NEO-Five Factor Inventory (NEO-FFI; *Costa & McCrae, 1992*). This self-report questionnaire consists of 60 items. The person has to respond whether he strongly disagrees, disagrees, hold a neutral position, agrees, of strongly agrees with each item. There are five factors: Neuroticism (*e.g.*, "I am seldom sad or depressed"), Extraversion (*e.g.*, "I really enjoy talking to people"), Openness (*e.g.*, "I often try new or foreign food"), Agreeableness (*e.g.*, "I would rather cooperate with others than compete with them"), and Conscientiousness (*e.g.*, "I am a productive person who always get the job done"). The instrument has good psychometric properties in the general population and in people after a stroke (*Dwan et al., 2017*). In this study we obtained a Cronbach's α of 0.83 for Neuroticism, 0.71 for Extraversion, 0.68 for Openness, 0.70 for Agreeableness, and 0.77 for Conscientiousness.

The Illness Cognitions Questionnaire (*Evers et al., 2001*; *Lauwerier et al., 2010*) was used to measure Acceptance (*e.g.*, "I have learned to live with the disease."). Other scales are the Helplessness-scale (*e.g.*, "My illness controls my life") and the Disease Benefits scale (*e.g.*, "My illness has helped me realise what is important in life."), but these scales were not used in this study. Each of the 18 items was to be scored on a four-point scale. Cronbach's α for the Acceptance-scale was 0.91 in this study.

The Dutch version of the five-item Satisfaction with Life Scale (*Diener et al., 1985*) was used to measure global life satisfaction (*e.g.*, "I am satisfied with my life"). Participants indicated how much they agree on a seven-point scale. We found a Cronbach's α of 0.80. The scores are summed up to a total score from 5 to 35. The European Brain Injury Questionnaire (EBIQ; *Teasdale et al., 1997*) is a measure of disease specific quality of life. It consists of 63 items, reflecting frequent occurring complaints after a brain injury, each of which are scored on a three-point-scale ("Not at all", "A little"; "A lot"). There are eight specific scales: (1) Somatic, (2) Cognitive, (3) Motivation, (4) Impulsivity, (5) Depression, (6) Isolation, (7) Physical, (8) Communication and one general Core scale, which consisted of those items with the highest communality in factor analysis (*e.g.*, "Problems in general."). The internal consistency and reliability of the original English questionnaire are sufficient (*Sopena et al., 2007*). *Van Bost, Van Damme & Crombez (2017)* made a Dutch translation of this questionnaire. In a Dutch-speaking sample they found a

Cronbach's α of 0.90 for the Core scale, which we used in this study. In the present study the Cronbach's α was 0.90.

Cognitive flexibility was measured with the paper version of Winsconsin Card Sorting Test (WCST; *Heaton et al., 1981*). People must match a card to one of four sample cards that vary in the colour (red, blue, yellow, green), form (triangle, circle, cross, star) of geometric figures, or the number of geometric figures (1–4) on the card. They are not given the sorting rule and have to find this through trial-and-error from feedback after each trial. Without warning, after ten consecutive correct answers the rule is changed. Participants have to complete six series or maximum 128 cards. Responses as a result of following the previous rule, instead of the actual rule, are called perseverative errors. The percentage of perseverative errors is an indication of the difficulty people have with flexible rule shifting.

## Procedure

Ethical approval was obtained from the Ethical Committee of the Faculty of Psychology and Educational Sciences of the Ghent University (2012/66). Written informed consent was obtained from the participants. They also gave permission to the rehabilitation professional of the service to give information about demographics and illness characteristics to the researcher.

An outline of the study procedure is shown in Fig. 1. Data were collected in the participants' therapeutic service, which in the majority of cases was a rehabilitation centre. The rehabilitation professional, a clinical psychologist or an occupational therapist, was asked to provide information about the demographics and the aetiology of the ABI. Rehabilitation professionals also provided expert ratings about the consequences of the injury. Assessment started with the EBIQ, because people were most familiar with reporting complaints, followed by the ICQ, the GAS and the SWLS, the NEO-FFI and the WCST. The ICQ was not included in the analyses. In most cases this was done in two sessions of 60 min. Some people needed three sessions, because the procedure was too demanding for them. The participants filled out the questionnaires in the presence of a researcher, who could help them to stay focused or could clarify the items of the questionnaire. Some people had difficulties reading the questions or ticking the right boxes. In these cases help was provided by the researcher.

## Statistical analysis

Descriptive statistics were provided for the demographic information, such as gender, age and education, and for time since injury, the expert ratings of the illness characteristics, personality factors, percentage perseverative answers, goal adjustment and QOL and life satisfaction. Pearson's product moment correlation analyses were used to examine strength and direction of linear association between study variables. We performed preliminary analyses to ensure no major violation of assumptions of normality, linearity, homoscedasticity, and multicollinearity in the main analyses. Then, we conducted four separate hierarchical multiple regression analyses.

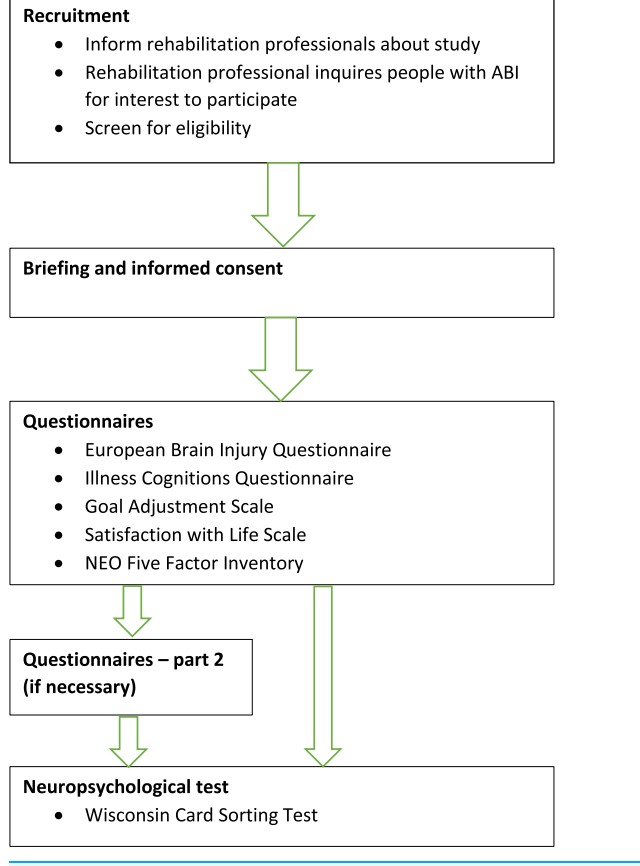

**Figure 1 Study procedure.**     

The first two analyses tested the effect of cognitive flexibility on goal disengagement and goal reengagement. The following predictors were entered: age, gender and education (Step 1), time since injury, cognitive impairments, self-awareness, communication impairments and motor impairments (Step 2), and percentage of perseverative errors (Step 3). The second two analyses tested whether goal adjustment has a predictive value for life satisfaction and QOL beyond personality. Predictors were entered in the following order: age, gender and education (Step 1), time since injury, cognitive impairments, self-awareness, communication impairments and motor impairments (Step 2), Neuroticism, Extraversion, Openness, Agreeableness and Conscientiousness (Step 3), and Goal Disengagement and Goal Reengagement (Step 4). To investigate whether there was a difference between the personality of this group of individuals with an ABI and that of the general population the average scores of the Big Five personality factors were compared to the norms of the Dutch version of the NEO-FFI (*Hoekstra, Ormel & de Fruyt, 2002*) using five one sample t-tests. All analyses were conducted in SPSS Version 27 using two-sided hypothesis testing with an alpha level of 0.05.

## RESULTS

Table 1 displays mean scores, standard deviations, and correlations. The percentage of perseverative responses is positively correlated to time since injury and negatively to goal

**Table 1 Intercorrelations between indicators of QOL, acceptance, goal adjustment and impairments.**

| Scale | Mean | SD | 1 | 2 | 3 | 4 | 5 | 6 | 7 | 8 | 9 | 10 | 11 | 12 | 13 | 14 | 15 | 16 | 17 |
|---|---|---|---|---|---|---|---|---|---|---|---|---|---|---|---|---|---|---|---|
| 1. Gender | | | 1 | – | – | – | – | – | – | – | – | – | – | – | – | – | – | – | – |
| 2. Education (years) | 12.76 | 2.69 | −0.26* | 1 | – | – | – | – | – | – | – | – | – | – | – | – | – | – | – |
| 3. Age | 44.38 | 14.50 | −0.06 | 0.06 | 1 | – | – | – | – | – | – | – | – | – | – | – | – | – | – |
| 4. Time since injury | 27.42 | 40.86 | 0.09 | −0.19 | −0.04 | 1 | – | – | – | – | – | – | – | – | – | – | – | – | – |
| 5. Cognitive problems | 3.12 | 1.17 | 0.01 | −0.22 | −0.31** | −0.21 | 1 | – | – | – | – | – | – | – | – | – | – | – | – |
| 6. Self-awareness | 5.45 | 1.39 | −0.01 | 0.04 | 0.27* | −0.02 | −0.57*** | 1 | – | – | – | – | – | – | – | – | – | – | – |
| 7. Commun problems | 2.17 | 1.26 | −0.04 | −0.06 | −0.19 | 0.23* | 0.11 | −0.22 | 1 | – | – | – | – | – | – | – | – | – | – |
| 8. Motor problems | 3.59 | 1.13 | 0.01 | −0.27 | −0.13 | 0.21 | 0.09 | −0.20 | 0.32** | 1 | – | – | – | – | – | – | – | – | – |
| 9. Neuroticism | 32.12 | 8.97 | −0.20 | 0.14 | 0.03 | 0.14 | 0.01 | 0.07 | 0.01 | −0.01 | 1 | – | – | – | – | – | – | – | – |
| 10. Extraversion | 40.03 | 6.54 | −0.03 | −0.04 | −0.15 | −0.06 | −0.00 | −0.11 | 0.01 | −0.07 | −0.46*** | 1 | – | – | – | – | – | – | – |
| 11. Openness | 35.26 | 6.62 | −0.30** | 0.47*** | 0.12 | −0.04 | 0.33* | 0.32** | −0.13 | −0.26* | 0.18 | 0.03 | 1 | – | – | – | – | – | – |
| 12. Agreeableness | 43.61 | 6.15 | −0.27* | −0.02 | 0.22 | −0.10 | −28 | 0.33** | 0.01 | −0.07 | −0.19 | 0.21 | −0.09 | 1 | – | – | – | – | – |
| 13. Conscientiousness | 44.86 | 7.16 | −0.08 | −0.17 | −0.06 | −0.08 | −90.05 | 0.02 | 0.22 | 0.05 | −0.36** | 0.39*** | −0.19 | 0.32** | 1 | – | – | – | – |
| 14. % perseverations | 24.88 | 21.71 | −0.05 | −0.19 | 0.19 | 0.40*** | −0.22 | −0.05 | 0.21 | 0.21 | 0.01 | −0.03 | −0.18 | 0.05 | −0.01 | 1 | – | – | – |
| 15. GAS Disengage | 10.53 | 3.61 | 0.18 | −0.15 | 0.20 | −0.09 | 0.05 | 0.04 | −0.01 | −0.12 | 0.04 | −0.08 | −0.14 | −0.10 | 0.00 | 0.09 | 1 | – | – |
| 16. GAS Reengage | 20.95 | 5.69 | −0.14 | 0.19 | 0.00 | −0.04 | −25* | 0.09 | −0.15 | −0.15 | −0.13 | 0.26* | 0.26* | −0.01 | 0.20 | −0.27* | 0.07 | 1 | – |
| 17. SWLS | 20.09 | 7.42 | 0.04 | 0.08 | 0.09 | −0.07 | −0.18 | −17 | −0.30** | −0.29** | −0.43*** | 0.40*** | 0.06 | −0.05 | 0.17 | −0.11 | −0.12 | 0.40*** | 1 |
| 18. EBIQ Core | 55.40 | 11.74 | −0.08 | 0.13 | 0.04 | 0.11 | 0.05 | 0.21 | 0.01 | −0.02 | 0.64*** | −0.47*** | 0.17 | −0.16 | −0.42*** | 0.09 | −0.04 | −0.34** | −0.54*** |

Notes:

SWLS, Satisfaction With Life Scale; EBIQ, European Brain Injury Questionnaire; GAS, Goal Adjustment Scale.
* $p < 0.05$.
** $p < 0.01$.
*** $p < 0.001$.

**Table 2 Mean scores, SD of the personality factors and comparison with the general population.**

| Personality factor | Average ABI group | SD ABI group | df | t | p | Cohen's d |
|---|---|---|---|---|---|---|
| Neuroticism | 32.12 | 8.97 | 73 | 0.98 | 0.330 | 0.114 |
| Extraversion | 40.03 | 6.54 | 73 | −0.10 | 0.924 | −0.011 |
| Openness | 35.26 | 6.62 | 73 | −0.84 | 0.406 | −0.097 |
| Agreeableness | 43.61 | 6.15 | 73 | −0.69 | 0.494 | −0.080 |
| Conscientiousness | 44.86 | 7.16 | 73 | −0.52 | 0.602 | −0.061 |

**Table 3 Hierarchical regression analysis of goal disengagement and goal reengagement.**

| | | Goal disengagement | | | | | Goal reengagement | | | | |
|---|---|---|---|---|---|---|---|---|---|---|---|
| Step | Predictors | β final (standardized) | Semi partial correlation | ΔF | ΔR² | Adj. R² | β final (standardized) | Semi partial correlation | ΔF | ΔR² | Adj. R² |
| 1 | Gender | 0.19 | 0.18 | 1.95 | 0.08 | 0.04 | −0.16 | −0.15 | 0.94 | 0.04 | −0.00 |
| | Age | 0.21 | 0.19 | | | | −0.00 | −0.00 | | | |
| | Education | −0.11 | −0.10 | | | | 0.07 | 0.06 | | | |
| 2 | Time since injury | −0.17 | −0.15 | 0.87 | 0.06 | 0.03 | 0.14 | 0.13 | 0.80 | 0.06 | −0.02 |
| | Cognitive problems | 0.16 | 0.13 | | | | −0.06 | −0.05 | | | |
| | Self-awareness | 0.04 | 0.03 | | | | 0.14 | 0.11 | | | |
| | Communicative problems | 0.07 | 0.06 | | | | −0.07 | −0.07 | | | |
| | Motor problems | −0.17 | −0.15 | | | | −0.02 | −0.02 | | | |
| 3 | % of perseverative errors | 0.11 | 0.10 | 0.66 | 0.01 | 0.03 | −0.28* | −0.25 | 4.27* | 0.06 | 0.03 |

**Note:**
* $p < 0.05$

reengagement. Life satisfaction and disease specific QOL correlated also with goal reengagement, but not with goal disengagement. Neuroticism and Extraversion are also related to life satisfaction and QOL, the latter also being related to Consciousness.

The average scores of the personality factors and the corresponding scores from the Dutch speaking reference group are presented in Table 2. We found no significant difference of the scores of the ABI group with those of the general population.

The results of a first set of hierarchical regression analyses are presented in Table 3. The percentage of perseverative responses predicted negatively goal reengagement beyond the previous steps ($\Delta F (1, 60) = 4.27$, $p = 0.043$), but had no effect on not goal disengagement ($\Delta F (1, 60) = 0.66$, $p = 0.421$).

Results from the second set of hierarchical regression analyses predicting disease specific QOL and life satisfaction are presented in Table 4. The impact of the ABI characteristics was significant for life satisfaction ($\Delta F (5, 65) = 4.27$, $p = 0.002$) and did not reach significance for QOL ($\Delta F (5, 65) = 2.29$, $p = 0.056$). The Big Five personality traits predicted life satisfaction ($\Delta F (5, 60) = 5.37$, $p < 0.001$) and disease specific QOL ($\Delta F (5, 60) = 9.10$, $p < 0.001$). Neuroticism was the only significant predictor of disease specific QOL and life satisfaction.

Goal adjustment was associated with life satisfaction ($\Delta F (2, 58) = 5.64$, $p = 0.006$) beyond the previous steps, but did not reach statistical significance for disease specific QOL ($\Delta F (2, 58) = 2.95$, $p = 0.060$). Goal reengagement was associated with QOL and life

Table 4 Hierarchical regression analysis (final model) of QOL and life satisfaction.

| Step | Predictors | EBIQ core | | | | | SWLS | | | | |
|---|---|---|---|---|---|---|---|---|---|---|---|
| | | β final (standardized) | Semi partial correlation | ΔF | ΔR² | Adj. R² | β final (standardized) | Semi partial correlation | ΔF | ΔR² | Adj. R² |
| 1 | Gender | −0.07 | −0.06 | 1.17 | 0.05 | 0.01 | 0.02 | 0.02 | 0.54 | 0.02 | −0.02 |
| | Age | −0.04 | −0.03 | | | | 0.20 | 0.18 | | | |
| | Education | 0.18 | 0.14 | | | | −0.09 | −0.07 | | | |
| 2 | Time since injury | 0.02 | 0.02 | 2.29 | 0.14 | 0.09 | 0.09 | 0.08 | 4.27** | 0.24 | 0.17 |
| | Cognitive problems | 0.27* | 0.20 | | | | −0.21 | −0.16 | | | |
| | Self-awareness | 0.32* | 0.21 | | | | −0.38** | −0.25 | | | |
| | Communicative problems | 0.03 | 0.03 | | | | −0.21** | −0.18 | | | |
| | Motor problems | 0.06 | 0.05 | | | | −0.31** | −0.26 | | | |
| 3 | Neuroticism | 0.44*** | 0.35 | 9.10*** | 0.35 | 0.44 | −0.34** | −0.28 | 5.37*** | 0.23 | 0.38 |
| | Extraversion | −0.09 | −0.07 | | | | 0.12 | 0.10 | | | |
| | Openness | −0.02 | −0.02 | | | | 0.01 | 0.01 | | | |
| | Agreeableness | −0.03 | −0.02 | | | | −0.18 | −0.14 | | | |
| | Conscientiousness | −0.14 | −0.11 | | | | 0.05 | 0.04 | | | |
| 4 | Disengagement | −0.03 | −0.03 | 0.04 | 0.04 | 0.47 | −0.22* | −0.20 | 5.64** | 0.08 | 0.37 |
| | Reengagement | −0.23* | −0.19 | | | | 0.29** | 0.25 | | | |

Notes:
EBIQ, European Brain Injury Questionnaire; SWLS, Satisfaction with Life Scale.
* $p < 0.05$.
** $p < 0.01$.
*** $p < 0.001$.

satisfaction. After controlling for the previous steps, goal disengagement only predicted life satisfaction.

## DISCUSSION

This study investigated whether goal adjustment abilities of individuals with an ABI were influenced by cognitive flexibility, which is often affected by brain injury. Previous research with this population (*Van Bost, Van Damme & Crombez, 2020*) has reported that goal reengagement was positively associated with quality of life and life satisfaction. In this study we investigated whether this effect is still present after controlling for personality.

The results can be readily summarized. First, cognitive flexibility was negatively associated with goal reengagement, but not with goal disengagement, even after controlling for demographic factors, time since injury and illness characteristics. Second, goal reengagement explained a unique portion of the variance of both QOL and life satisfaction after controlling for demographic factors, time since injury, severity and the Big Five personality traits. Third, goal disengagement was only negatively associated with life satisfaction. Fourth, neuroticism was the only personality factor predicting both QOL and life satisfaction.

The impact of goal disengagement and goal reengagement on a large number of outcomes is well established (*Barlow, Wrosch & McGrath, 2020*). Less is known about the factors contributing to people's ability to a flexible goal adjustment. The ability to

flexibly change one's behaviour as a response to a changing situation may be a prerequisite for goal adjustment. This is in particular relevant for people with an ABI, because a brain injury often results in a reduction of cognitive flexibility. Our measure of cognitive flexibility, the percentage of perseverative errors on the WCST, did explain an additional part of the variance of goal reengagement, but not goal disengagement. As far as we know this is the first study investigating the impact of executive functioning on goal adjustment. The effect of cognitive flexibility was significant beyond the illness characteristics, estimated by the therapist of the participant. It is therefore unlikely that these results could be attributed to a more general impairment. Goal reengagement may require a certain level of cognitive functions as divergent thinking and concept formation (*Drago & Heilman, 2012*). Interestingly, recent research has shown that the results on the WCST are not only related to cognitive flexibility but also to cognitive persistence or the tendency to put effort in cognitive demanding tasts (*Teubner-Rhodes et al., 2017*). People with a higher tendency to put effort in these cognitive tasks may also be more inclined towards putting effort in the search for new life goals and engagement towards them.

The finding that goal reengagement explained a unique proportion of the variance of both indicators of QOL corroborates the conclusions of earlier research with people with an ABI (*Van Bost, Van Damme & Crombez, 2020*). It is also in concordance with an overall positive relationship between goal reengagement and well-being in long-term health conditions, reported by *Scobbie et al. (2020)*. In their review mixed results for goal disengagement were reported. A majority of the studies included in this review found a positive relationship with indicators of wellbeing. In our study we found no relationship of goal disengagement with disease specific QOL and a negative relationship with life satisfaction. This may be surprising, because goal disengagement could help to avoid confrontation with repeated failures and may free up resources for other goals (*Wrosch et al., 2003*). *Barlow, Wrosch & McGrath (2020)* concluded in their meta-analysis that goal disengagement has a negative association with negative indicators of wellbeing, such as anxiety or negative affect, but not with positive indicators, such as life satisfaction or purpose in life. Goal reengagement was negatively associated with the negative indicators, as well as positively with the positive indicators. This is in contradiction with our results, because goal disengagement was negatively related to life satisfaction, a positive indicator of wellbeing, and not with the EBIQ, essentially a list of possible negative consequencens of an ABI. Nevertheless, others had found similar results in patients with chronic pain (*Esteve et al., 2018*). Their interpretation was that people understood that they had to abandon cherished life goals, but they did so with frustration and distress. In people with multiple sclerosis a high goal disengagement, in combination with a low goal reengagement, was even related to more depression (*Van Damme et al., 2019*).

The reason why there is a negative effect of goal disengagement on life satisfaction may relate to the content of particular items of the SWLS. Two of the five items of the SWLS require the respondents to reflect on their life as a goal-oriented project ("So far I have gotten the important things I want in life.", "If I could live my life over, I would change almost nothing."). People with a high score on disengagement report that they easily let

go of goals. This may lead to less accomplishment and therefore less satisfaction with what they realised in life. As a follow up of this interpretation, we performed a *post hoc* analysis of the effect of goal disengagement on those two items. We found that goal disengagement was negatively related to those two items, but not to the remaining three. After an ABI people often realise that they have no other choice than to disengage from their previous goals. However, this absence of goals to strive for may lead to a sense of emptiness.

We found no differences on the Big Five personality factors between our sample of people with an ABI and the general population. Other studies, with different designs and using information of a significant other (*Leonardt, Schmukle & Exner, 2016*; *Norup & Mortensen, 2015*; *Lannoo et al., 1997*) reported a decline in extraversion and conscientiousness following an ABI. It is not possible to give an unambiguous interpretation of our results. Nevertheless, we found no evidence for such a personality change. As expected (*Schimmack et al., 2004*; *Dwan & Ownsworth, 2017*), neuroticism had an important negative impact on QOL and life satisfaction. This was not unexpected, given the conclusions of the review of *Dwan & Ownsworth (2017)* in individuals with a stroke. None of the other personality factors yielded a significant result on life satisfaction or on the disease specific QOL.

This study has some limitations. First, this study has a cross-sectional design, making causal interpretations impossible. Second, sample size was small, as a result of which the study might have been underpowered to detect smaller effects. Third, the age range was broad. As can be seen in Table 1, there is a negative correlation between age and cognitive problems. This may be surprising, but it may relate to differences in etiology. In older participants the cause of the ABI was usually stroke, leading to a more discrete impairment, often in the domain of motor or communication functions. In younger participants, the cause of the ABI is usually a TBI, leading to a broader and more cognitive symptomatology. Fourth, most measures were based on self-report measures. It could be argued that people with an ABI, due to their cognitive deficits, rely more on their pre-injury self-image than on their analysis of the present situation. Moreover, for some of our respondents it was difficult to use abstract and generalized concepts about their personality and goal adjustment strategies. Fifth, cognitive flexibility was assessed only by means of the Wisconsin Card Sorting Test.

Our study has clinical implications, in particular our findings about goal reengagement. It means that helping patients with an ABI reengage towards new goals is useful, regardless of their personality profile. Rehabilitation professionals have to take into account that it may be more challenging for people with an ABI to reengage towards new goals, especially if they suffer from deficits in cognitive flexibility. This may require standard procedures in the rehabilitation plan to test the executive functioning of the patients. In the work with people with cognitive impairments it is always necessary to make specific adaptations to therapeutic interventions (*Gallagher, McLeod & McMillan, 2016*). People with problems in cognitive flexibility may need more active guidance to reengage themselves towards more feasible life goals. Rehabilitation professionals may present them with alternative options or they can stimulate them to try out new activities and interests.

## CONCLUSION

Striving towards personal goals is important for wellbeing. When these goals are no longer achievable as a result of an ABI, people that reengage in other, more feasible goals reported more life satisfaction and quality of life. However, goal reengagement is also a cognitive process. This study showed that the cognitive symptoms of an ABI could hamper this adjustment process. We also observed an effect of goal reengagement and to a lesser extent of goal disengagement on wellbeing beyond that of the Big 5 personality traits.

Future studies are necessary to better understand the precise nature of the cognitive difficulties leading to problems in goal reengagement. Further research also needs to explore if people with an ABI use different forms of goal adjustment based on goal characteristics such as the importance or attainability of these goals.

### Funding

The authors received no funding for this work.

### Competing Interests

The authors declare that they have no competing interests.

### Author Contributions

- Gunther Van Bost conceived and designed the experiments, performed the experiments, analyzed the data, prepared figures and/or tables, authored or reviewed drafts of the article, recruiting participants, and approved the final draft.
- Stefaan Van Damme analyzed the data, authored or reviewed drafts of the article, and approved the final draft.
- Geert Crombez conceived and designed the experiments, analyzed the data, authored or reviewed drafts of the article, and approved the final draft.

### Human Ethics

The following information was supplied relating to ethical approvals (*i.e.*, approving body and any reference numbers):

The Ethical Commission of the Faculty of Psychology and Educational Sciences - Ghent University grants a favourable opinion to the project (2012/66).

### Data Availability

The raw results are available in the Supplemental File.

### Supplemental Information

Supplemental information for this article can be found online at http://dx.doi.org/10.7717/peerj.13531#supplemental-information.

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
