# Peer review of "Goal adjustment and well-being after an acquired brain injury: the role of cognitive flexibility and personality traits"

_PeerJ, doi:10.7717/peerj.13531_

## Round 0.1 · original submission · Minor Revisions

Dear Authors,

Please attend to the minor revisions required as per the comments of the 3 peer reviewers.

Reviewer 1 ·

Basic reporting

It is a well-structured written manuscript with a clear and unambiguous statement and relevant literature were included. The figures and tables were appropriately included. No major comment on the basic reporting.

Experimental design

The overall design of the study is appropriate, with sufficient data included. All the tools used were clearly mentioned and the details procedures were clearly described. However, there are a few comments regarding the design.

Page 13, lines 183 – 184; I would suggest if the author could include one figure to illustrate the procedure.

Validity of the findings

The results were well reported and discussed. However, there are some issues with regards to the finding specifically in terms of the participants involved and maybe could be discussed:

Page 10, lines 114-115; Participants’ age ranged from 19 to 66 years. Looking at the wide range of the participant's age, I am afraid it might affect the overall finding. I would suggest a much more focused age group. Even though the author did not discuss this variable, but from the finding, it is clear that there is a correlation between age and cognitive problems in general. How do you see this variable in understanding your finding as age is known to affect cognitive ability and the rehabilitation process?

Page 10, lines 115-116; some of the participants were of unique characteristics such as having stroke and history of brain surgery. how do you see this dynamic influencing your result?

Additional comments

no additional comment

·

Basic reporting

Needs small improvements.

Experimental design

Needs small improvements.

Validity of the findings

Findings are valid. Reporting needs small improvements.

Additional comments

Thanks for opportunity to review manuscript entitled ‘‘Goal adjustment and well-being after an acquired brain injury: the role of cognitive flexibility and personality traits’’ for Peerj Journal. The author/authors examined the association between goal adjustment and well-being among acquired brain injury individuals. The strength of the manuscript includes examining variables of interest in a sample where such studies are scarce, very good use of English, Overall, the article is generally very well written and deserves to be published in this journal. However, some minor revisions must be made before the publication of the article. Because my main philosophy of reviewing a manuscript as reviewer and sometimes an editor to improve the manuscript and not punishing the authors, I provided very specific and detailed peer review of the manuscript to increase its quality and citation potential. I hope authors of the manuscript may benefit from my review. Minor revisions reported section by section with the page and line number and when possible with suggestions.
Minor Revisions
Title
1. Title, Page 4, Line Number, Not available: The title of manuscript must be revised for correctness. The correct form of title is ‘ ‘Goal adjustment and well-being after an acquired brain injury: The role of cognitive flexibility and personality traits’’.
Abstract
2. Abstract, Page 4, Line Number, Not available : ‘ ‘78 persons with an ABI completed a set of questionnaires.’’ Must be corrected. It is incorrect to use numbers at the beginning of sentence as per APA 7 rules. The correct form is ‘ ‘Seventy eight persons with an ABI completed a set of questionnaires.’’
3. Abstract, Page 4, Line Number, Not available : Please revise following sentence ‘ ‘Indicators of wellbeing were the European Brain Injury Questionnaire (EBIQ) and the Satisfaction with Life Scale (SWLS).’’ One revision may be that ‘ ‘Indicators of wellbeing were measured with the European Brain Injury Questionnaire (EBIQ) and the Satisfaction with Life Scale (SWLS).’’
4. Abstract, Page 4, Line Number, Not available : Authors must provide long name of the NEO-FFI (NEO Five-Factor Inventory) in its first use in Abstract. Moreover, authors must used statistical analyses to Method section of abstract.
5. Abstract, Page 4, Line Number, Not available : No need to following sentence in Abstract and must remove from the manuscript ‘ ‘…..i.e., goal disengagement and goal reengagement’’.
6. Keywords are completely missing in the manuscript and must be added.
Introduction
7. Introduction, Page 5, Line 4: Please revise following sentence for correctness ‘ ‘Head traumata and stroke are main causes.’’ One revision may be that ‘ ‘Head trauma and stroke are main causes.’’ Moreover, This sentence is very assertive. Thus, add a citation to this sentence.
8. Introduction, Page 5, Line 4-5: The citation/citations need for following sentence ‘ ‘Less common causes are anoxia, brain tumours, substance abuse and brain infections.’’
9. Introduction, Line 12-15: Following sentences are unrelated o content of manuscript and must remove from manuscript. ‘ ‘Nevertheless, accepting of the reality of the problem has been shown to improve the subjective quality of life (Van Bost, Van Damme & Crombez, 2017). Of importance, acceptance is not simply surrendering to an unwanted reality. It is ”recognizing the need to adapt to chronic illness while perceiving the ability to tolerate the unpredictable, uncontrollable nature of the disease and handle its adverse consequences” (Evers, 2001, p. 1027). Diller (2005) considers acceptance to be a marker of mental health and a desirable outcome of rehabilitation efforts.’’
10. Introduction General: Apart from above correction Introduction is very well written. I congratulate the authors. I really very much liked the section Line 24-47.
Method
11. Method, Page 9 Line, 103: 68 participants must be 78 participants.
12. Method, Page 9 Line, 107: participating centres must be participating centers.
13. Method, Page 9 Line, 114: M and SD must be italic.
14. Method, Page 9 Line, 117: SD must be italic.
15. Method, Page 9 Line, 118: M and SD must be italic.
16. Method, Page 9 Line, 120-123: Following sentences must move to Procedure section ‘ ‘Ethical approval was obtained from the Ethical Committee of the Faculty of Psychology and
Educational Sciences of the Ghent University (2012/66). Written informed consent was obtained from the participants. They also gave permission to the rehabilitation professional of the service to give information about demographics and illness characteristics to the researcher.’’
17. Method, Page 11, Line 155: Please revise following sentence for clarity ‘ ‘Cronbach’s α for the Acceptance-scale was .91’’ One revision may be that ‘ ‘Cronbach’s α for the Acceptance-scale was .91 in this study’’.
18. Method, Page 11, Line 156-158: Please revise following sentence ‘ ‘The Dutch version of the 5-item Satisfaction with Life Scale (Diener, Emmons, Larssen, & Griffin, 1983) measures judgements about global life satisfaction (e.g. “I am satisfied with my life”).’’ One revision may be that‘ ‘The Dutch version of the 5-item Satisfaction with Life Scale (Diener, Emmons, Larssen, & Griffin, 1983) was used to measure global life satisfaction (e.g. “I am satisfied with my life”).’’
19. Method, Page 11, Line 166-167: Please revise the following sentence. It is very difficult to understand ‘ ‘The psychometric properties are sufficient (Sopena, Dewar, Nannery, Teasdale, & Wilson, 2007). Van Bost, Van Damme, & Crombez (2017) made a Dutch translation.’’
20. Method, Page 11, Line 168: Please revise the following sentence ‘ ‘ …… which will also be used in this study’’ One revision may be that ‘ ‘…..which we used in this study’’.
21. Method, Page 11, Line 168: Authors must add Core’s internal consistency coefficient calculated in their study (Cronbach’s alpha). Moreover, authors must add possible scores for missing scales. For example Core.
22. Method, Page 11, Line 172: Please revise the following sentence ‘ ‘ …… or number (1-4).’’ One revision may be that ‘ or number in this test’’. I am not able to understand what authors want to mean with ‘‘(1-4)’’
23. Method, Page 13, Line 185: The reason why authors did not used the ICQ in their analyses are unclear and authors must give information about reasons. Moreover, ıf they did not used this scale, I think authors must completely remove from Measures section but not sure about this. The last decision related to this pertaining to editor.
24. Method, Page 13, Line 191: Please add such as after the demographic information the new form looks like this ‘ ‘…….demographic information such as time since injury, the expert ratings of the illness characteristics….’’
25. Method, Page 13, Line 193-194: Please revise the following sentences ‘ ‘Associations between those variables were examined using Pearson correlation coefficients.’’ A better writing may be that ‘ ‘We used Pearson’s product moment correlation analyses to examine strength and direction of linear association between study variables.’’ or ‘ ‘Pearson’s product moment correlation analyses were used to examine strength and direction of linear association between study variables.’’
26. Method, Page 13, Line 195: Please revise the following sentences ‘‘Two hierarchical multiple regression analyses were then conducted.’’ One revision may be that ‘ ‘Then, we conducted two separate two hierarchical multiple regression analyses.’’
27. Method, Analysis: Authors must change Analyses title as Statistical Analyses. Moreover, author must add brief information about assumption testing in regression and correlation analyses. For example, ‘ ‘Preliminary analyses conducted to ensure no major violation of assumptions of normality, linearity, homoscedasticity, and multicollinearity in the main analyses.’’ or ‘‘We performed preliminary analyses to ensure no major violation of assumptions of normality, linearity, homoscedasticity, and multicollinearity in the main analyses.’’ Homoscedasticity of variance in multiple regression, and homogeneity of variance in t-test is the same thing. Thus, writing one of them is enough. Some researchers unfamiliar with general linear models thinks they are different.
28. Method, Analysis: As a multivariate analyst, I can confidently say that unpaired t-tests are completely wrong if authors compared to difference between the personality of this group of individuals with an ABI and that of the general population. The true analysis is one sample t-tests if standard deviation is unavailable. If standard deviation available for population one sample z-test must be used. However, standard deviation is an approximation in one sample z-test if sample size is no fairly large. Thus, one sample t-test is a better choose in most situations. Moreover, I calculated one sample t-test for randomly chosen variables from Table 2 and found similar results. authors may be wrongly reported their used analyses.
29. Method, Analysis: As a multivariate analyst, I think information/writing about hierarchical multiple regression analyses are very repetitive. Moreover, authors did not conduct two separate regression analyses, rather they used four separate regression analyses. Two for different goals , one for life satisfaction, and one for quality of life. Apart from above corrections, Method section is well written.
Results
30. Results, General: Authors must report all findings with two decimal except for p value (three decimal) in the Tables.
31. Results, General: Authors must add Notes. Under the Tables before reporting significance levels.
32. Results, General: Authors must add commas between significance level under the tables.
33. Results, Table 2: Authors must add Cohen d effect size estimates after they are correcting the analyses.
34. Results, Table 3, Table 4: Authors must add semi-partial correlations to Table 3, and Table in the final model after the β final (standardized).
35. Results, Table 4: Abbreviations must be under the table in Table 4.
36. Results, General: Authors need to correct reporting all F change reporting. Add a space between to degrees of freedoms, and p value reporting and change with delta. Wrong: (F change (1,60) = 4.27, p=.043). True: (ΔF (1, 60) = 4.27, p = .043). Apart from above corrections, Result section is well written.
Discussion
37. Discussion, Page 16, Line 251-252: Authors must remove following sentence. I think it is unnecessary. ‘ ‘as measured by the percentage perseverative errors on the WCST.’’.
38. Discussion, Page 16, Line 267: Authors must remove following sentence. I think it is unnecessary. ‘ ‘i.c. cognitive flexibility,’’
39. . Discussion, Page 19, Line 324-128: Although I read six times fourth limitation , I am not able to understand what I have to conclude. Authors need to completely revise following part ‘ ‘It is not possible to make strong statements based on a single test. The choice for the full version of the WCST prevented us to test for cognitive persisitence. Teubner-Rhodes et al. (2017) developed a measure of cognitive persistence based on the short version of the Wisconsin Card Sorting Test.’’
40. Discussion, Clinical Implications: Clinical implications are written for others who may benefit from research findings. Thus, as a writing tip don’ use any time we statement in this section otherwise very necessary. Always use such as mental health professionals, practitioner, researchers, clinicians, counselors, rehabilitation professionals. Thus, I strongly recommend to rewrite sentence with we in this section.
41. Discussion, General: Conclusion section and future studies is completely missing in this section. Author must add a paragraph after clinical implications section and must add a conclusion paragraph and recommendations for future studies.
42. Apart from above corrections, Discussion section is well written.

Reviewer 3 ·

Basic reporting

Basic reporting is adequate.

Experimental design

Experimental design is adequate and true.

Validity of the findings

The findings are valid.

Additional comments

Thanks for giving me the opportunity to review the manuscript entitled ‘‘Goal adjustment and well-being after an acquired brain injury: the role of cognitive flexibility and personality traits’’ for Peerj Journal. The author/authors examined the association between goal adjustment and well-being among acquired brain injury individuals. The strength of the manuscript includes examining variables of interest in a sample where such studies are not common. Overall, the article is very well written and deserves to be published in this journal. However, some minor revisions must be made before the publication of the article. I hope authors of the manuscript may benefit from my review. Minor revisions reported section by section with the page and line number and when possible with suggestions.
Minor Revisions
Title
1. Title, Page 4, Line Number, Not available: The title of manuscript must be revised for correctness. The correct form of title is ‘‘Goal adjustment and well-being after an acquired brain injury: The role of cognitive flexibility and personality traits’’.
Abstract
2. Abstract, Page 4, Line Number, Not available : ‘ ‘78 persons with an ABI completed a set of questionnaires.’’ Should be changed as ‘ ‘Seventy eight persons with an ABI completed a set of questionnaires.’’ in terms of APA 7.

Introduction
3. Introduction, Page 5, Line 3: Please add another sentence between first and second sentence. There is a sharp pass between them. One revision may be that ‘ ‘Although there are various causes for it, traumata and stroke are main causes.’’

Method
4. Method, Page 9 Line, 103: 68 participants must be 78 participants.
5. Method, Page 9 Line, 107: participating centres must be participating centers.
6. Method, Page 9 Line, 114: M and SD must be italic.
7. Method, Page 9 Line, 117: SD must be italic.
8. Method, Page 9 Line, 118: M and SD must be italic.

Results
9. Results, General: Authors must report all findings with two decimal except for p value (three decimal) in the Tables.
10. Results, General: Authors must add Notes. Under the Tables before reporting significance levels.
11. Results, General: Authors must add commas between significance level under the tables.
12. Results, Line 227, F change should not be reported like.

Discussion
12. In discussion section implications might be strengthened for clinicians and researchers.

References
13. References, Please correct the uppercase letters for this reference ‘Brandtstädter, J., & Rothermund, K. (2002). The Life-Course Dynamics of Goal Pursuit and 349 Goal Adjustment: A Two-Process Framework. Developmental Review, 22(1), 117-150.

14. There are same mistakes through the reference list. İt should be checked.

---

## Round 0.2 · accepted · Accept

Thank you. Your manuscript has been accepted.

Reviewer 1 ·

Basic reporting

no comment

Experimental design

no comment

Validity of the findings

no comment

Additional comments

all the comments and suggestions have been taken into consideration and improvements to the manuscript have been made.

·

Basic reporting

The manuscript is clear and unambiguous, professional English used throughout.

Experimental design

Rigorous investigation performed to a high technical & ethical standard.

Validity of the findings

All underlying data have been provided; they are robust, statistically sound, & controlled.

Additional comments

Thanks for opportunity review revised manuscript entitled ‘‘Goal adjustment and well-being after an acquired brain injury: the role of cognitive flexibility and personality traits’’. I would like the thanks to authors. They make a good job for improving quality of their manuscript. Authors revised the manuscript as I requested with a good will. In this form, Introduction reflects very well the previous studies and study aim, Method section and Result section is correct, and Discussion section adequately synthesis to previous study findings and current study results. Overall, I have no further comment regarding to manuscript. I congratulate to authors and wish them success on their future endeavors.

Reviewer 3 ·

Basic reporting

It is suitable.

Experimental design

It is suitable.

Validity of the findings

The validity of the findings is correct.